# DONOD: Efficient and Generalizable Instruction Fine-Tuning for LLMs via Model-Intrinsic Data Selection

## Abstract

Ad-hoc instruction fine-tuning of large language models (LLMs) is widely adopted for domain-specific adaptation. While domain-specific supervised fine-tuning (SFT) is effective and efficient, it often weakens cross-domain generalization and struggles with noisy training data. To address these challenges, we propose DONOD, a lightweight model-intrinsic data selection method. Our approach evaluates data using two model-parameter-based metrics: Delta of Norm (DON), which captures the cumulative influence on model weights, and Norm of Delta (NOD), which quantifies weight instability. Moreover, by employing the Technique for Order of Preference by Similarity to Ideal Solution (TOPSIS) algorithm, we effectively filter noisy, unlearnable, and generalization-harming samples without relying on auxiliary models during the SFT process. Experiments on mathematical tasks demonstrate that data selected by DONOD achieves superior fine-tuning efficiency and improved robustness against noisy data. By filtering out 70% of the whole dataset, we improve target-domain accuracy by 14.90% and cross-domain accuracy by 5.67%. Meanwhile, our selected data present superior cross-architecture generalization. Data pruned by smaller models (e.g., Llama 3.1-8B) generalize effectively on larger models (e.g., Llama 2-13B). Compared to existing related methodologies, DONOD demonstrates comparable or superior performance while remaining dataset-agnostic, enabling broader applicability. Code will be made publicly available soon.

## 1 Introduction

In recent years, large language models (LLMs) have demonstrated strong generalization capabilities and remarkable success across a wide range of applications (Achiam et al., 2023; Meta AI, 2024; Yang et al., 2025a; Bai et al., 2025). While foundation models pretrained on massive corpora provide a powerful starting point, effectively adapting them to specific user needs or domain-specific tasks often requires fine-tuning. In practice, many real-world scenarios demand rapid, on-demand adaptation, a process we refer to as ad-hoc instruction fine-tuning. Unlike universal instruction fine-tuning, which aims for broad capability, ad-hoc fine-tuning focuses on a specific set of task instructions to enhance a particular ability or domain knowledge. This approach assumes

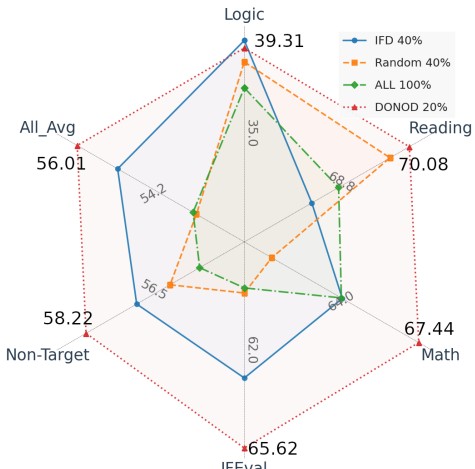

Figure 1: Performance of DONOD across various benchmarks. Our 20% selected dataset outperforms or matches the full-data training baseline in most evaluation dimensions. This demonstrates that DONOD enables efficient fine-tuning with significantly fewer training samples, while improving generalization performance.

the base model is already instruction-aligned (e.g., an Instruct-type model) and prioritizes efficient specialization. This setting is particularly relevant for startups, researchers, and domain experts who need to customize LLMs for narrow applications, such as adding a language, creating internal tools, powering customer support, or enabling domain-specific reasoning. While still potentially consuming a significant number of tokens, the process is focused on a restricted domain.

However, ad-hoc fine-tuning of LLMs with massive instruction datasets incurs substantial computational costs. At scale, these training costs are tremendous and often unaffordable for startups or researchers (Xia et al., 2022; Yang et al., 2025c). For instance, fine-tuning a 13B-parameter model on hundreds of millions of instruction–response pairs can require thousands of GPU hours. Even more concerning, recent studies indicate that the quality of fine-tuning data is more critical than its quantity (Li et al., 2025b; Xia et al., 2024; Wang et al., 2023). This is because large-scale instruction data collected via web scraping or weak supervision often contain substantial noise and redundancy (Li & Zhang, 2021; Szep et al., 2024; Yang et al., 2024). Consequently, significant resources may be wasted on examples that contribute little to, or even degrade, the final model's performance.

To overcome these challenges, data selection has been proposed as a promising solution. By identifying a compact yet highly representative subset from existing SFT data (Li et al., 2025b; Xia et al., 2024), it is possible to retain or even improve model performance compared to full-dataset training, while reducing training costs. Many existing methods, reward-model-based filtering (Xu et al., 2025; Yang et al., 2025b) or gradient-based selection (Xia et al., 2024), have achieved promising results in accelerating training and improving the efficiency of the LLM fine-tuning process. Despite the promising results, many of these methods incur huge computational overhead or rely on task-specific validation sets (Xia et al., 2024; Xie et al., 2023), which may limit their scalability across diverse domains. Furthermore, recent studies have found that models fine-tuned on such data are prone to domain overfitting along with degraded generalization across domains (Li & Zhang, 2021; Szep et al., 2024). This limited cross-domain generalization poses a unique challenge for ad-hoc instruction fine-tuning, as we aim to improve performance on the target domain without compromising general capabilities in other domains. These issues highlight the urgent need for a principled, data-centric approach that can accelerate instruction fine-tuning by reducing training overhead while preserving generalization, thereby enabling more efficient, scalable, and robust LLMs training. This raises a central question: *How can we select the most representative samples from large-scale datasets to enable efficient, generalizable, and robust fine-tuning of LLMs?*

To address the challenges, we propose **DONOD**, a model-intrinsic data selection method that identifies a compact yet highly informative subset of training data points. Specifically, DONOD introduces two complementary metrics derived from the model's training dynamics: Delta of Norm (DON) and Norm of Delta (NOD), as detailed in Section 3.2. To reconcile these dual objectives, maximizing generalization via DON while minimizing harmful fluctuations via NOD, we adopt the Technique for Order of Preference by Similarity to the Ideal Solution (TOPSIS) algorithm (Hwang & Yoon, 1981; Chakraborty, 2022) to rank samples based on their proximity to the ideal selection criterion. Importantly, DONOD requires no auxiliary models, domain-specific heuristics, or validation sets. It leverages only intrinsic training signals, enabling scalable, efficient, and fully self-supervised selection. Extensive experiment results across diverse benchmarks and LLM architectures show that DONOD achieves training acceleration while preserving or even exceeding the full-data generalization performance with significantly fewer training examples. For instance, compared with the full-data SFT setting, DONOD achieves a 14.90% gain in target-domain accuracy and a 5.67% gain in cross-domain accuracy using only 30% of the data. Furthermore, our method shows strong cross-architecture generalization, consistently performing well on models of varying structures and scales. Since most existing methods are not robust to more complex and realistic noisy settings, we further validate the robustness of DONOD in more challenging scenes, highlighting its practical significance.

The contributions can be summarized as follows: **(1)** We propose DONOD, a lightweight and model-intrinsic data selection framework for fine-tuning acceleration of LLMs, significantly reducing training costs while maintaining performance. **(2)** We propose two complementary metrics, DON and NOD, to jointly ensure the generalization of the selected samples and reduce noisy or unlearnable samples. **(3)** Extensive experiments across diverse benchmarks and architectures demonstrate the superior fine-tuning performance, particularly in cross-domain and cross-architecture generalization, highlighting the method's practicality for scalable and robust LLM training.

## 2 RELATED WORK

Data selection for supervised fine-tuning on LLMs critically impacts model performance. Traditional methods often rely on external models as quality judges ((Du et al., 2023), (Chen et al., 2024)) or employ reward models to identify high-quality data (Yang et al., 2025b). However, this dependence on auxiliary models, e.g., trained from scratch, incurs significant computational costs and limits scalability. Recent studies ((Li et al., 2025a)) further question the effectiveness of this paradigm.

Alternative approaches focus on intrinsic data metrics. For instance, Cao et al. (2024) proposes evaluating data quality through features like length, naturalness, and coherence. However, the field lacks consensus on universal metrics: while Chen et al. (2023) emphasizes diversity, Liu et al. (2024) argues for prioritizing complex or challenging samples. This ambiguity motivates the third category, model-intrinsic methods. These methods leverage the model's training dynamics to bypass explicit metric definitions. As noted by Jiang et al. (2019); Yang et al. (2025c), the model's response to data inherently signals its utility for learning, enabling automated data selection.

Model-intrinsic selection methods branch out based on various scenarios. The first category assumes access to a target data distribution, often via validation or development sets. For example, Mindermann et al. (2022) approximates loss differences between holdout and training sets, while Xia et al. (2024) uses gradient similarity between validation and training data. These methods falter when target distributions are ambiguous or undefined, which is common in scenarios aiming to enhance broad capabilities rather than optimize for specific benchmarks.

The second category eliminates reliance on target distributions. Works like (Wang et al., 2024), (Jiang et al., 2019), (Loshchilov & Hutter, 2016), and (Li et al., 2024b) employ loss or perplexity thresholds, assuming high-loss samples are valuable learning challenges. However, this assumption proves brittle for noisy or mislabeled data (Yang et al., 2024), where high loss reflects annotation errors rather than learnable patterns. Furthermore, challenging samples may exceed the model's current capacity, rendering them unproductive for training.

Both categories neglect cross-domain generalization. Methods targeting specific distributions risk catastrophic forgetting, where performance gains on target tasks degrade generalizability. Conversely, loss-based selection exacerbates this by prioritizing samples that induce significant weight updates, destabilizing pre-trained knowledge.

To address these limitations, we propose DONOD. DON functions as a proxy of generalization, while NOD recognizes that the sample causes significant instability in the model weight. Integrated via the TOPSIS, DONOD filters noisy, unlearnable, and generalization-harming samples without auxiliary models or predefined targets.

## 3 THE PROPOSED METHOD

### 3.1 OVERVIEW

Our proposed method is summarized in Figure 2. The approach consists of three core components: 1) DON and NOD metrics based on the Frobenius norm are used to estimate the samples' impact on model weight update. 2) TOPSIS is a multi-objective decision mechanism that balances task-specific gain and cross-domain generalization, ensuring that the selected subset preserves both in-domain effectiveness and robustness to distribution shifts. 3) By approximating the full model behavior through changes in the output layer, DONOD enables lightweight and scalable selection without the need to backpropagate through the entire model. These components construct an efficient data selection framework that supports accurate, low-cost subset selection, enabling fine-tuning with significantly fewer samples while maintaining or even improving model performance.

### 3.2 DON AND NOD METRICS

Let $D$ denote an ad-hoc dataset for instruction fine-tuning. For a specific data sample $D_i \in D$, let $\{W^l\}_l^L$ represent the weight of the model before fine-tuning and $\{W'^l\}_l^L$ the weight matrix after fine-tuning on $D_i$. We employ the DON and NOD to quantify this change. Specifically, the DON is

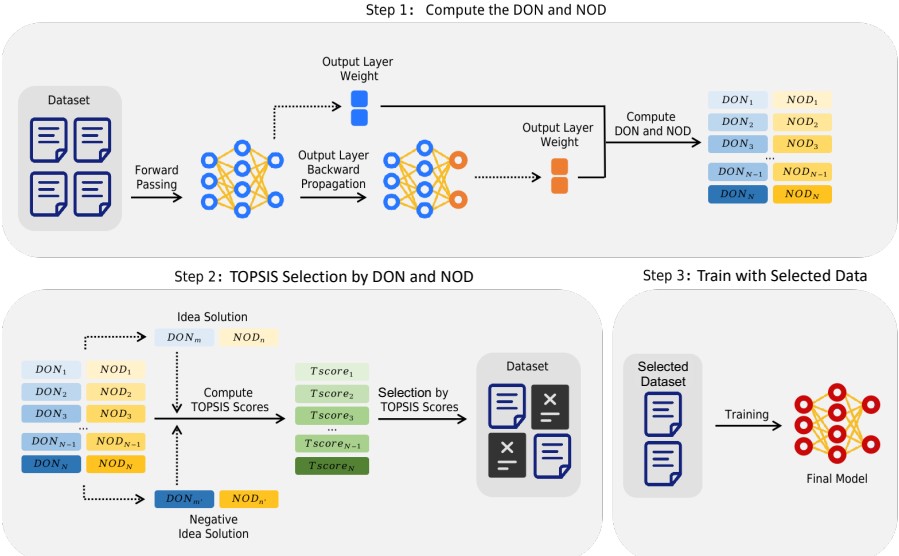

Figure 2: Overview of our proposed DONOD, which follows a lightweight pipeline: (1) Compute DON and NOD metrics for each sample and (2) Apply TOPSIS to select representative data and filter harmful/low-quality data.

defined as:

$$\text{DON} = \sum_{l=1}^{L} \left( \|W^l\|_F - \|W'^l\|_F \right) = \sum_{l=1}^{L} \left( \sqrt{\sum_{i=1}^{m_l} \sum_{j=1}^{n_l} |w_{i,j}^l|^2} - \sqrt{\sum_{i=1}^{m_l} \sum_{j=1}^{n_l} |w_{i,j}'^l|^2} \right), \quad (1)$$

where $m_l$ and $n_l$ are the dimensions of the weight matrix $W^l$ of layer $l$, $\|\cdot\|_F$ denotes the Frobenius norm. Here, we adopt the Frobenius norm due to its ability to capture fine-grained structural changes across all weight elements while maintaining computational efficiency, making it a suitable and scalable proxy for quantifying sample-level influence in large-scale models. Thus, DON captures the cumulative shift in the model's weight magnitude. From a generalization perspective, a positive DON suggests that a sample reduces the model's Frobenius norm, which is associated with lower complexity and better generalization (Bartlett, 1996; Yin et al., 2020; Shalev-Shwartz & Ben-David, 2014). A detailed theoretical justification for this connection is provided in Appendix B and C. Meanwhile, the NOD is defined as:

$$\text{NOD} = \sum_{l=1}^{L} \|W^l - W'^l\|_F = \sum_{l=1}^{L} \sqrt{\sum_{i=1}^{m_l} \sum_{j=1}^{n_l} \|w_{i,j}^l - w_{i,j}'^l\|^2}, \quad (2)$$

which measures the direct geometric displacement of weights in parameter space at the current step. Thus, in the context of SFT, NOD quantifies how drastically a single sample perturbs the parameter space. These metrics are complementary: DON reflects the overall scaling of weights, whereas NOD reflects the sensitivity of model weight on a single sample.

While the Frobenius norm can be applied to whole model weights, prior works (Nadipalli, 2025; Rosati et al., 2024) show that fine-tuning primarily affects later layers, with the output layer acting as a bottleneck for domain adaptation. Therefore, we estimate the sample influence using the weights of the last layer, which brings two benefits: 1) Computational Efficiency: The output layer is typically smaller than the hidden layers, reducing the computational cost of computing norms across iterations, 2) Interpretability: Output layer updates correlate more directly with task performance, avoiding the entangled representations of deeper layers.

In this way, we derive a simplified version of the computation of Eq.1 and Eq.2:

$$\text{DON} = \|W^L\|_F - \|W'^L\|_F = \sqrt{\sum_{i=1}^{m_L} \sum_{j=1}^{n_L} |w_{i,j}^L|^2} - \sqrt{\sum_{i=1}^{m_L} \sum_{j=1}^{n_L} |w_{i,j}'^L|^2}, \quad (3)$$

where $m_L$ and $n_L$ are the dimensions of the output layer.

$$\text{NOD} = \|W^L - W'^L\|_F = \sqrt{\sum_{i=1}^{m_L} \sum_{j=1}^{n_L} |w_{i,j}^L - w_{i,j}'^L|^2}. \tag{4}$$

## 3.3 INTEGRATION OF TOPSIS

TOPSIS is a multi-criteria decision analysis (MCDA) method that ranks alternatives by their relative closeness to an ideal solution. In DONOD, TOPSIS is employed to resolve the inherent tension between the two metrics, DON and NOD, by identifying samples that simultaneously maximize DON (to enhance generalization) and minimize NOD (to avoid noise). After computing the DON and NOD for each sample, we apply TOPSIS to rank the data points.

TOPSIS inherently balances conflicting objectives, i.e., maximizing DON and minimizing NOD, by leveraging geometric distance in the normalized metric space. Moreover, normalization mitigates the impact of differing metric magnitudes, ensuring neither DON nor NOD dominates the ranking. It avoids subjective weight assignment (unlike weighted sum) and provides a total ordering of samples (unlike Pareto optimality), which is critical for deterministic selection decisions. Thus, we choose TOPSIS in our framework. Details of the algorithm are provided in Appendix F.

## 3.4 COMPUTATIONAL COMPLEXITY

The algorithm's computational complexity consists of: (1) per-sample DON and NOD computation and (2) TOPSIS-based sample selection.

In the first phase, we perform a forward pass ($O(P)$ time per sample, where $P$ is the model's parameter count), a backward pass restricted to the output layer ($O(O)$ time, $O$ being the output layer's parameters), and compute Frobenius norms for weight updates ($O(O)$). Since $O \subset P$, the per-sample cost simplifies to $O(P)$, resulting in a total training complexity of $O(N \cdot P)$, where $N$ is the number of training samples.

The second phase involves normalizing DON/NOD metrics ($O(N)$), computing distances to ideal and negative-ideal solutions ($O(N)$), and ranking samples via TOPSIS scores, dominated by an $O(N \log N)$ sorting step. Thus, the selection process has a total complexity of $O(N \log N)$. Combining both phases, the dominant term is $\mathcal{O}(N \cdot P)$, as $P \gg N \log N$ in modern neural networks (e.g., $P \sim 10^6$–$10^{12}$ parameters). The overall time complexity simplifies to $\boxed{\mathcal{O}(N \cdot P)}$. For storage, only $O(N)$ space is required to store per-sample metrics, as no intermediate model states need to be retained. Therefore, the runtime of DONOD can be approximated by the model's inference speed. In our experiments using Llama-3.1-8B-Instruct, processing the SAT Math COT dataset took approximately 18 minutes of wall-clock time on a single A100 80GB GPU, with negligible storage requirements.

## 4 EXPERIMENT

### 4.1 EXPERIMENT SETUP

**Evaluation Benchmarks and Training Datasets** Following Ma et al. (2025), we construct our evaluation benchmark based on AGIEval (Zhong et al., 2024) and IFEval (Zhou et al., 2023). This benchmark is designed to be comprehensive and domain-orthogonal, assessing abilities in logical reasoning, mathematics, reading comprehension, and instruction following. By incorporating diverse datasets, the benchmark reflects real-world ad-hoc SFT scenarios, where the objective is to strengthen targeted model abilities rather than optimize for narrow or unrepresentative benchmarks.

To comprehensively evaluate DONOD's effectiveness across diverse domains, tasks, and data conditions, we select data that span a wide range of settings, including variations in domain (e.g., mathematics, logical reasoning, instruction following), task format (e.g., chain-of-thought, multiple-choice, fine-grained evaluation), and the presence or absence of validation sets. This design enables robust benchmarking under realistic and varied constraints. Specifically, we assess DONOD across

Table 1: Experimental results with domain-specific averages on LogiQA Train and GSM8K. The **Non-Target** column shows average performance excluding logic reasoning or Math (target domain). The **All Avg** column shows the average of all tasks in the benchmark. All values are percentages.

| Dataset | Method | Logic↑ | Reading↑ | Math↑ | IFEval↑ | Non-Target↑ | All Avg↑ |
|---------|--------|--------|----------|-------|---------|-------------|----------|
| **LogiQA Train** | LESS 5% | 22.62 | **25.19** | 21.47 | 64.33 | 36.99 | 27.91 |
| | Random 5% | 20.45 | 24.27 | 24.06 | 46.95 | 31.96 | 26.35 |
| | ALL 100% | **22.98** | 24.02 | 22.26 | 17.74 | 21.34 | 22.77 |
| | DONOD 5% | 20.91 | 24.27 | **24.22** | 68.95 | 39.15 | 28.86 |
| **GSM8K** | ALL (100%) | 33.43 | 70.74 | **47.99** | 59.70 | 48.96 | **48.74** |
| | LESS 5% | 34.74 | 71.85 | 25.78 | 60.63 | 50.46 | 44.97 |
| | DONOD 5% | **37.16** | **71.92** | 33.69 | **71.16** | **52.74** | 48.51 |

Table 2: Experimental results with domain-specific averages on SAT Math COT and IFEval-Like Data. The **Non-Target** column shows average performance excluding mathematical reasoning or IFEval (target domain), revealing how methods generalize to other abilities.

| Dataset | Method | Logic↑ | Reading↑ | Math↑ | IFEval↑ | Non-Target↑ | All Avg↑ |
|---------|--------|--------|----------|-------|---------|-------------|----------|
| **SAT Math COT** | IFD 40% | 39.31 | 68.26 | 64.18 | 63.03 | 56.87 | 55.04 |
| | ALL 100% | 37.12 | 68.76 | 64.14 | 59.70 | 55.19 | 53.23 |
| | Random 40% | 38.32 | 69.72 | 61.17 | 59.89 | 55.98 | 53.15 |
| | DONOD 30% | **39.98** | 67.62 | **73.70** | 63.40 | 57.00 | **56.25** |
| | DONOD 20% | 38.96 | **70.08** | 67.44 | **65.62** | **58.22** | 56.01 |
| **IFEval-Like Data** | ALL (100%) | 36.06 | 64.92 | 44.84 | 64.33 | 48.60 | 48.68 |
| | IFD 40% | 34.06 | **66.31** | 34.04 | **71.90** | 44.80 | 46.55 |
| | DONOD 30% | **36.14** | 57.82 | **66.47** | 46.77 | **53.47** | **49.01** |

the 4 settings, **SAT Math Chain-of-Thought (COT)** (Davidson, 2023) (math, COT, no validation set), **LogiQA-Train** (Liu et al., 2020) (logical reasoning, multiple-choice, with validation set), **IFEval-like Data** (Xu et al., 2024) (instruction-following, general, no validation set) and **GSM8K** (Cobbe et al., 2021) (math, COT, biased distribution vs. SAT Math and Aqua-RAT).

**Models and Experiment Settings**  We evaluate DONOD on a diverse instruction-tuned models to assess its generalizability across architectures and fine-tuning paradigms. Specifically, we consider: (1) LLaMA-3.2-3B-Instruct (Meta AI, 2024), a lightweight model optimized for instruction-following tasks; (2) LLaMA-3.1-8B-Instruct (Grattafiori et al., 2024), a mid-sized model widely used in recent instruction-tuning studies; (3) LLaMA-2-13B-Chat (Touvron et al., 2023), a larger model trained with conversational objectives; and (4) Qwen 2.5-7B-Instruct (Team, 2024), a model from a distinct architecture family, differing in tokenizer, training data, and parameterization. This ensures a comprehensive evaluation of DONOD under varied model designs and training strategies. We focus on the output layer of the Llama-3.1-8B-Instruct model for our experiment.

## 4.2 COMPARISON WITH STATE-OF-THE-ARTS

As shown in Table 1 and Table 2, DONOD consistently outperforms other baseline methods while using less data across nearly all benchmarks. Specifically, our method achieves the best performance in core reasoning tasks such as math and logic, outperforming full-data baselines in both target-domain accuracy and cross-domain generalization. For instance, on GSM8K, DONOD with only 5% of the data achieves higher logic, reading, and IFEval scores than training with 100% of the data, while on SAT Math COT, DONOD with 20–30% data yields notable improvements over full-data fine-tuning in both math reasoning and overall averages. Notably, it exhibits strong cross-task and cross-domain transferability without relying on task-specific tuning or heuristics, and remains competitive even in challenging settings such as reading comprehension. The strong gains in the Non-Target and All Avg columns further highlight that DONOD not only strengthens task-specific reasoning but also transfers well to broader abilities such as instruction following. These results demonstrate that, during the LLM fine-tuning process, DONOD achieves training acceleration using substantially less training data, offering a scalable and generalizable solution for data-efficient LLM fine-tuning.

Table 3: Cross-architecture generalization of selecting data with Llama-3.1-8B-Instruct and fine-tuning on Llama-2-13b-chat, Qwen-2.5-7B-Instruct, and Llama-3.2-3B-Instruct.

| Method | Logic (↑) | Reading (↑) | Math (↑) | IFEval (↑) | Non-Target (↑) | All Avg (↑) |
|---|---|---|---|---|---|---|
| Llama-2-13b-chat | | | | | | |
| ALL (100%) | 30.57 | 54.68 | 23.38 | 29.94 | 38.39 | 35.28 |
| DONOD 20% | 31.52 | 55.58 | 26.89 | 29.39 | 38.83 | 36.57 |
| DONOD 30% | 31.30 | 54.61 | 23.39 | 29.57 | 38.49 | 35.53 |
| Qwen-2.5-7B-Instruct | | | | | | |
| ALL (100%) | 43.62 | 73.99 | 74.57 | 56.01 | 57.87 | 58.90 |
| DONOD 20% | 41.95 | 68.28 | 79.86 | 58.23 | 56.15 | 59.09 |
| DONOD 30% | 42.57 | 71.15 | 77.07 | 61.74 | 58.48 | 59.33 |
| Llama-3.2-3B-Instruct | | | | | | |
| ALL (100%) | 9.23 | 34.67 | 10.60 | 63.96 | 35.95 | 23.10 |
| DONOD 20% | 27.65 | 50.46 | 34.78 | 65.80 | 47.97 | 38.71 |
| DONOD 30% | 29.46 | 56.75 | 33.92 | 65.06 | 50.42 | 41.03 |

Table 4: Experimental results with domain-specific averages. The **Non-Target** column shows average performance (%) excluding mathematical reasoning (target domain), revealing how methods generalize to other abilities.

| Method | Logic↑ | Reading↑ | Math↑ | IFEval↑ | Non-Target↑ | All Avg↑ |
|---|---|---|---|---|---|---|
| DON | 36.19 | 68.12 | 58.15 | **66.17** | 56.83 | 52.38 |
| NOD | **40.23** | 67.12 | 65.71 | 58.04 | 55.13 | 54.39 |
| Weighted Sum | 38.08 | 69.15 | 63.74 | 53.79 | 53.67 | 53.54 |
| Pareto Optimization | 36.85 | 70.02 | 57.82 | 65.25 | 57.37 | 53.07 |
| DONOD | 38.96 | **70.08** | **67.44** | 65.62 | **58.22** | **56.01** |

## 4.3 CROSS-ARCHITECTURE GENERALIZATION

To assess the cross-architecture generalization of our selected data points, we select data points using Llama-3.1-8B-Instruct and then fine-tune the datasets using Llama-2-13b-chat, Qwen-2.5-7B-Instruct, and Llama-3.2-3B-Instruct. As shown in Table 3, our selected subsets (e.g., 20% and 30%) not only retain but also surpass the full-data baseline in overall performance across diverse models. Notably, DONOD yields consistent gains in overall averages, with particularly large improvements for the smaller Llama-3.2-3B-Instruct, where 20–30% subsets boost the average score by more than 15 points over the full-data performance. These results suggest that, despite differences in size and architecture, LLMs share a consistent perception of instruction difficulty. In (Li et al., 2024a), this consistency is demonstrated through metrics like perplexity and Instruction-Following Difficulty scores, which show strong rank correlations across models of different sizes. As a result, smaller models like GPT-2 can effectively filter instruction data for much larger models, such as LLaMA2-7B or GPT-4. DONOD builds on the same principle, but instead of external metrics, it leverages intrinsic parameter-level signals to identify universally useful samples. In practice, this means that data selected with a smaller model (e.g., Llama-3.1-8B) can transfer effectively to larger or structurally different models (e.g., Llama-2-13B or Qwen-2.5-7B), echoing the weak-to-strong transfer effect. These results highlight the strong cross-architecture transferability of DONOD-selected samples, underscoring the practical utility for scalable and data-efficient fine-tuning across heterogeneous model families.

## 4.4 ROBUSTNESS IN IDENTIFYING NOISE

Real-world datasets often involve noise, where mislabeled or poorly aligned samples can degrade model performance and hinder generalization. Unfortunately, creating clean and diverse datasets is time-consuming and expensive. Therefore, it is necessary to evaluate the robustness of data selection methods under noisy settings. In this study, we conduct a controlled experiment on the SAT Math CoT dataset using the LLaMA 3.1 model. We start with the top 20% of samples originally selected by DONOD. To simulate real-world data imperfections, we introduce controlled noise by randomly masking words in the labels of these clean samples, which mimics subtle corruption or annotation errors. These perturbed samples are then reintegrated into the full dataset, creating a new training

pool containing embedded noisy instances. We reapply DONOD to this dataset to select a new top 20% subset based on the updated DON and NOD values. We assess sensitivity to noise by measuring the overlap between the original and newly selected top 20%. The result shows a drop to only 38.7% overlap, indicating that DONOD successfully identifies and filters out many of the newly corrupted samples. This experiment highlights DONOD's strong responsiveness to fine-grained label corruption and its ability to dynamically adapt selection criteria.

## 4.5 ABLATION STUDY AND ANALYTICAL RESULTS

To understand the effect of each component in DONOD, we conduct an ablation study on the SAT Math CoT dataset using LLaMA-3.1-8B-Instruct. As summarized in Table 4, we evaluate four configurations: (1) ranking by DON only, (2) ranking by NOD only, (3) joint usage of DON and NOD without TOPSIS (via weighted sum or Pareto Front), and (4) the full DONOD method (DON + NOD + TOPSIS). We do not consider configurations such as DON + TOPSIS, since the TOPSIS framework inherently requires multiple criteria to balance conflicting signals.

**Effect of Individual Metrics**  When applied individually, the two metrics exhibit complementary behaviors. NOD achieves stronger performance on the target domain (e.g., Math: 65.71%), as it emphasizes samples that induce substantial localized parameter updates, thereby favoring task-specific adaptation. However, this comes at the cost of reduced generalization to non-target domains, suggesting susceptibility to overfitting. In contrast, DON promotes smoother and more stable parameter updates, which better preserve generalizable knowledge. This results in superior cross-domain generalization but comparatively weaker gains in task-specific reasoning. These findings confirm that DON and NOD capture distinct yet complementary aspects of sample importance.

**Combination Strategies**  Directly combining DON and NOD, such as a weighted sum or a Pareto Front, fails to fully reconcile their competing objectives. Weighted sum marginally improves reading comprehension but reduces math and non-target performance compared to using DON or NOD in isolation. Pareto Front, on the other hand, places greater emphasis on cross-domain stability but sacrifices task-specific accuracy. These results underscore the need for a principled integration mechanism to balance stability and specificity.

**Full Method (DONOD)**  The proposed DONOD achieves the best balance between domain-specific performance and cross-domain generalization. By ranking samples according to their proximity to the ideal trade-off between DON and NOD, TOPSIS provides a principled means of resolving conflicts between the two criteria. This yields improvements in both target-domain accuracy and transferability, while also enhancing the overall average performance. Moreover, these results highlight the necessity of integrating both DON and NOD within a multi-objective optimization framework, validating the effectiveness of TOPSIS.

**Validation of Output Layer Focus**  To further justify our design choice in Section 3.2, we analyze the sensitivity of weight changes across layers. By ranking the Frobenius norm delta across layers after fine-tuning Llama-2-13B-Chat on the SAT Math COT dataset, as shown in Figure 3, we observe that the output layer exhibits the largest shifts, reflecting its heightened responsiveness to task-specific supervision. Restricting DON and NOD to this layer thus provides a representative signal of overall weight dynamics, closely aligned with the layer-wise average Frobenius norm, while substantially reducing computational cost. Compared to computing Eq.1 and Eq.2 across all layers for every data sample, our approach only requires backpropagation through the final layer together with constant-time DON/NOD computations, yielding a highly efficient yet effective approximation.

**Stability of DONOD Across Data Proportions**  To evaluate the stability of DONOD under varying data regimes, we train models on subsets ranging from 10% to 100% of the data. As shown in Figure 4, DONOD exhibits consistent performance and remarkable efficiency, often matching or surpassing the full-data baseline with substantially fewer samples. Notably, with only 10% of the data, it outperforms the full-data baseline in both Logic and Overall Average, underscoring its robustness under data scarcity. At 20%, DONOD achieves an optimal trade-off, reaching peak scores in Logic and Reading while maintaining strong overall averages. These results demonstrate that DONOD

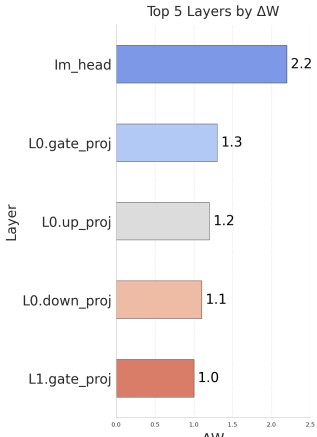

Figure 3: Rank of the top 5 layers by Frobenius norm delta after fine-tuning Llama-2-13B.

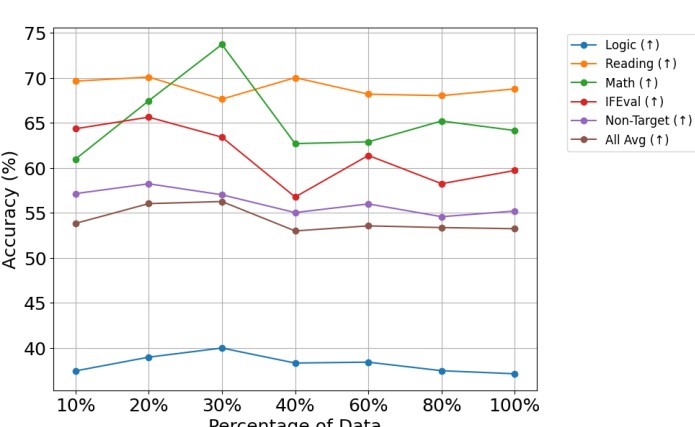

Figure 4: Stability of DONOD across data proportions.

reliably identifies high-quality samples that maximize informative learning signals while minimizing noise.

**Human Evaluation on Selected Datasets**
To further validate the effectiveness of our selection strategy, we conduct a human evaluation by inverting the process: instead of retaining the top-ranked samples, we keep those with the lowest TOPSIS scores, forming the *NODON* dataset. As shown in Figure 5, these samples cluster on the high-NOD, low-DON region of the NOD–DON plane. Manual inspection of the NODON-pruned SAT Math COT datasets reveals several recurring categories: (1) over-elaborated answers to trivial questions, (2) incomplete or partial responses, (3) incorrect reasoning steps, (4) mislabeled or flawed questions, and (5) overly complex or impractically difficult problems. These findings indicate that

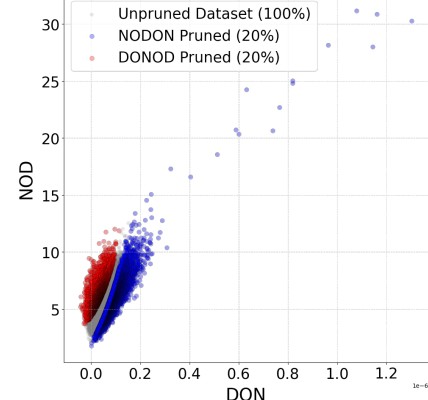

Figure 5: Illustration of the distribution of pruned dataset with DONOD and NODON (keep samples with the lowest TOPSIS scores, efficiently sample with high NOD and low DON).

DONOD systematically removes unhelpful or misleading samples, thereby improving instruction data quality and enhancing both the robustness and generalization of fine-tuned LLMs.

## 5 CONCLUSION

In this paper, we propose DONOD, a model-intrinsic data selection framework to enhance LLM fine-tuning efficiency without sacrificing model performance. By leveraging weight dynamics, our method selects high-quality data and suppresses the selection of noisy or uninformative data via dual complementary metrics, DON and NOD. These metrics are integrated via TOPSIS, enabling a principled trade-off between maximizing generalization and minimizing harmful updates. Experiments show that DONOD can reduce training data volume by up to 70% while outperforming standard supervised fine-tuning, achieving superior training acceleration. Notably, datasets selected by smaller models also generalize well when used to fine-tune other LLM architectures, underscoring the framework's scalability and practicality for real-world LLM pipelines. We hope DONOD inspires further research on data selection for LLM training from a model-intrinsic perspective and believe our method will serve as a promising dataset optimization tool for the community, enabling enhanced data-centric LLM training pipelines.

## 6 ETHICS AND REPRODUCIBILITY STATEMENT

The datasets (benchmarks) used for the evaluation and comparison of our method and baselines are publicly accessible, ensuring the transparency and reproducibility of our work. We will release our work to the community as soon as it is accepted, ensuring that our work is reproduced and grounded for other researchers and practitioners.

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

APPENDIX

## A  AI ASSISTANT USAGE STATEMENT

Large Language Models (LLMs) were employed in a limited capacity during the preparation of this paper, primarily for text refinement. Importantly, LLMs were not involved in formulating the core ideas, designing the methodology, or determining the structure and substantive content of the work.

## B  THEORETICAL FOUNDATIONS

The generalization gap for a learning algorithm, for a specific function $f$, is the difference between its true (expected) risk $R_\mathcal{D}(f)$ over the data distribution $\mathcal{D}$ and its empirical risk $R_S(f)$ over a finite sample $S = \{(x_1, y_1), \ldots, (x_m, y_m)\}$. A smaller generalization gap means the model's performance on unseen data is closer to its performance on training data, i.e., the model generalizes well.

The Rademacher complexity $R_m(\mathcal{F})$ of a hypothesis class $\mathcal{F}$ measures its ability to fit random noise. It is formally defined ((Bartlett & Mendelson, 2003) and (Neyshabur et al., 2015)) as:

$$R_m(\mathcal{F}) = \mathbb{E}_{\boldsymbol{\xi} \in \{\pm 1\}^m} \left[ \frac{1}{m} \sup_{f \in \mathcal{F}} \left| \sum_{i=1}^{m} \xi_i f(x_i) \right| \right]$$

Similarly by (Bartlett & Mendelson, 2003), for a hypothesis class $\mathcal{F}$ of real-valued functions and a 1-Lipschitz loss function $\ell$, a standard generalization bound, if inputs are bounded and output is bounded, states that for any $f \in \mathcal{F}$, with probability at least $1 - \delta$ over the random draw of the sample $S$:

$$R_\mathcal{D}(f) \leq R_S(f) + 2R_m(\mathcal{F}) + C\sqrt{\frac{\log(1/\delta)}{m}}$$

where $C$ is a constant related to the range of the loss function, and for 1-Lipschitz losses, it simplifies to $2R_m(\mathcal{F}) + \sqrt{\frac{\log(1/\delta)}{2m}}$. This inequality implies that the generalization gap $R_\mathcal{D}(f) - R_S(f)$ is bounded by terms that include the Rademacher complexity of the hypothesis class $\mathcal{F}$. Therefore, to show that the generalization gap of $M'$ is smaller, we need to show that the Rademacher complexity of its corresponding hypothesis class is smaller.

## C  THEORETICAL ANALYSIS

Here, based on our case, we simplify our network as a network with a fixed architecture ($d$ layers, width $H$) and RELU activations, and consider a hypothesis class of functions whose weights $w$ satisfy a bound on $\mu_{p,q}(w)$. Specifically, we are interested in the Frobenius norm, which is $\mu_{2,2}(w)$. Let $\mathcal{F}_\mu = N_{d,H,\sigma_{RELU}}^{\mu_{2,2} \leq \mu}$ be the class of functions that can be realized by such a network where the overall $\ell_2$ norm of its weights is at most $\mu$.

For the model $M_{high}$ of weight $W_{high}$, its actual Frobenius norm is $\mu_{M_{high}} = \mu_{2,2}(W_{high})$. The function $f_{W_{high}}$ computed by $M_{high}$ belongs to the hypothesis class $\mathcal{F}_{\mu_{M_{high}}}$.

Similarly, the network $M_{low}$ has weights $W_{low}$, and its actual Frobenius norm is $\mu_{M_{low}} = \mu_{2,2}(W_{low})$. The function $f_{W_{low}}$ computed by $M_{low}$ belongs to the hypothesis class $\mathcal{F}_{\mu_{M_{low}}}$. We are given $\mu_{M_{low}} > \mu_{M_{high}}$.

Here, we theoretically prove that models with high DON generalize better.

To resonate the use of DON, we first introduce two key mathematical tools, the generalization gap and Rademacher complexity, as shown in Appendix 1.1. Let $M$, $M_{high}$ and $M_{low}$ be three neural networks with identical architectures but distinct weight matrices $W$, $W_{low}$ and $W_{high}$, respectively. Here $M_{high}$ stands for a model with high DON, $M_{low}$ stands for a model with low DON, and $M$ is an arbitrary auxiliary model.

We show that the model with higher DON generalizes better, e.g., if

$$\text{DON}^{\text{high}} - \text{DON}^{\text{low}} = \sum_{l=1}^{L} \left( \|W^l\|_F - \|W^l_{\text{high}}\|_F \right)$$
$$- \sum_{l=1}^{L} \left( \|W^l\|_F - \|W^l_{\text{low}}\|_F \right) > 0 \tag{5}$$

then $M_{high}$ generalize better.

Simplify 5, we have:

$$\sum_{l=1}^{L} (\|W^l_{low}\|_F - \|W^l_{high}\|_F) > 0 \tag{6}$$

Using the same notation as (Neyshabur et al., 2015), let $\mu_{2,2}(W_{low}) = \sum_{l=1}^{L} \|W^l_{low}\|_F$ and $\mu_{2,2}(W_{high}) = \sum_{l=1}^{L} \|W^l_{high}\|_F$, suppose the Frobenius norms of the weights satisfy $\mu_{2,2}(W_{low}) > \mu_{2,2}(W_{high})$, we want to proof that that the generalization gap of $M_{high}$ is smaller than that of $M_{low}$.

**Proof.** Firstly, we make necessary assumptions and the setup based on our practical conditions as detailed in Appendix 1.2. According to the Corollary 2 of (Neyshabur et al., 2015), for any $d \geq 1$, $1 \leq p < \infty$, and $1 \leq q \leq p^* = p/(p-1)$ (where $1/p + 1/p^* = 1$), the Rademacher complexity of the class $N_{\mu_{2,2} \leq \mu}^{d, H, \sigma_{RELU}}$ is bounded.

In our case, we are considering the Frobenius norm, so $p = 2$ and $q = 2$. This means $p^* = 2/(2-1) = 2$. Since $q = 2$ and $p^* = 2$, the condition $q \leq p^*$ ($2 \leq 2$) is met. Therefore, we can apply the bound from Corollary 2:

$$R_m(N_{\mu_{2,2} \leq \mu}^{d, H, \sigma_{RELU}}) \leq \left( \frac{2\mu}{\sqrt[2]{d}} \right)^d R_{m,2,D}^{linear}$$

Here, $R_{m,2,D}^{linear}$ is the Rademacher complexity of $D$-dimensional linear predictors with unit $\ell_2$ norm with respect to a set of $m$ samples. Since $p = 2$, by the same Corollary 2 (Neyshabur et al., 2015), we have a bound for this term: $R_{m,2,D}^{linear} \leq \sqrt{\frac{\min\{p^*, 4log(2D)\} \max_i \|x_i\|_{p^*}^2}{m}}$. Let $K = \frac{2^d}{d^{d/2}} \sqrt{\frac{\min\{p^*, 4log(2D)\} \max_i \|x_i\|_{p^*}^2}{m}}$. This $K$ is a positive constant that depends only on the fixed architecture ($d$), input dimensionality ($D$), sample size ($m$), and the maximum $\ell_2$ norm of input data points (which is assumed finite).

So, the Rademacher complexity bound for our class becomes:

$$R_m(N_{d, H, \sigma_{RELU}}^{\mu_{2,2} \leq \mu}) \leq K \cdot \mu^d$$

For network $M_{high}$, its function $f_{W_{high}}$ belongs to the class $\mathcal{F}_{\mu_{M_{high}}}$, and its Rademacher complexity is bounded by:

$$R_m(\mathcal{F}_{\mu_{M_{high}}}) \leq K \cdot \mu_{M_{high}}^d$$

For network $M_{low}$, its function $f_{W_{low}}$ belongs to the class $\mathcal{F}_{\mu_{M_{low}}}$, and its Rademacher complexity is bounded by:

$$R_m(\mathcal{F}_{\mu_{M_{low}}}) \leq K \cdot \mu_{M_{low}}^d$$

As explained in Appendix 1.2, since we are given $\mu_{M_{low}} > \mu_{M_{high}}$, and $\mu_M, \mu_{M'}$ are non-negative (being norms), it directly follows that $\mu_{M_{low}}^d > \mu_{M_{high}}^d$. Therefore:

$$R_m(\mathcal{F}_{\mu_{M_{high}}}) \leq K \cdot \mu_{M_{high}}^d < K \cdot \mu_{M_{low}}^d \tag{7}$$

This shows that the Rademacher complexity of the hypothesis class associated with $M_{high}$ is strictly smaller than that associated with $M_{low}$.

From the generalization bound established in Appendix B: For $M_{high}$, with probability at least $1 - \delta$:

$$R_{\mathcal{D}}(f_{W_{high}}) - R_S(f_{W_{high}}) \leq 2R_m(\mathcal{F}_{\mu_{M_{high}}}) + C\sqrt{\frac{\log(1/\delta)}{m}}$$

For network $M_low$, with probability at least $1 - \delta$:

$$R_{\mathcal{D}}(f_{W_{low}}) - R_S(f_{W_{low}}) \leq 2R_m(\mathcal{F}_{\mu_{M_{low}}}) + C\sqrt{\frac{\log(1/\delta)}{m}}$$

Since Eq.7, it implies that the upper bound on the generalization gap for $M_{high}$ is lower than that for $M_{low}$, that is, in the worst case, the generalization gap $M_{high}$ can have is strictly less than $M_{low}$. In other words, $M_{high}$ is generally better generalized. We also provide a more intuitive interpretation of DON and NOD metrics in Appendix D.

## D  INTUITION BEHIND DON AND NOD METRICS

**DON as a Proxy for Generalization:** A negative DON indicates that fine-tuning on $D_i$ increases the model's weight magnitude. As shown above, we have proved that the model with high DON lead to better generalization. This is also supported by the study of (Shalev-Shwartz & Ben-David, 2014), its increment relates to the regularization principles, where smaller norms often correlate with lower generalization error (e.g., weight decay). Intuitively, samples that induce a significant increase in the Frobenius norm contribute to the complexity of the model, potentially damage its ability to generalize across domains, and high and positive DON indicate the simplification of the model and better generalization. Our experimental results support this intuition, showing that samples yielding high DON positive values improve cross-domain accuracy.

**NOD as an Indicator of Bad Sample:** A high NOD value indicates that the data point $D_i$ has a significant influence on the model. In the context of ad-hoc SFT, training begins with a model that already possesses a certain level of generalization. Since the model is unlikely to encounter entirely new information or learn fundamentally new concepts after pretraining, data points with high NOD values are often indicative of low-quality samples, such as mislabeled or noisy data. Therefore, our method focuses on filtering out samples with high NOD values, thereby removing noisy or unlearnable data from the training process.

## E  CHOICE OF FROBENIUS NORM

The Frobenius norm is a matrix norm defined for a matrix $W \in \mathbb{R}^{m \times n}$ as the square root of the sum of the squares of its elements, i.e., $\|W\|_F = \sqrt{\sum_{i=1}^{m} \sum_{j=1}^{n} |w_{i,j}|^2}$. Compared to other norms, it offers specific advantages. For instance, the $\ell_1$ norm is given by $\|W\|_1 = \sum_{i=1}^{m} \sum_{j=1}^{n} |w_{i,j}|$, it treats the matrix as a flattened vector and measures the total absolute deviation. The $\ell_1$ norm's robustness to outliers makes it suitable for measuring aggregate influence. However, it lacks sensitivity to the fine-grained linear transformation differences represented by the weight matrix, which is critical for data selection. Moreover, it provides a less effective measure of the magnitude difference for fine-grained data selection. To capture the slightest difference between samples, the Frobenius norm shows better performance. In terms of the $\ell_2$ norm, for a matrix, it typically implies the Spectral norm and is given by $\|W\|_2 = \sigma_{max}(W)$, where $\sigma_{max}(W)$ is the maximum singular value obtained from the singular value decomposition (SVD) of $W$. This makes the $\ell_2$ norm solely focus on the largest singular value, capturing the dominant direction of the matrix's transformation but ignoring the contribution of smaller singular values. This ignorance of finer structural changes in the weight matrix, making it less suitable for detecting sample-wise influences. Furthermore, the computation of SVD for a large matrix, which is a common situation for modern LLMs, can be a heavy workload, hindering the scalability of the method. For the same matrix $W \in \mathbb{R}^{m \times n}$, comparing with the expensive computation of $\ell_2$ spectral norm ($O(min(mn^2, m^2n))$), Frobenius norm is much more efficient and only requiring $O(mn)$.

## F  IMPLEMENTATION OF TOPSIS

1. Normalization: DON and NOD are normalized to eliminate scale differences. Given a matrix $W \in \mathbb{R}^{n \times 2}$, where $n$ is the number of samples and columns represent DON and

NOD, vector normalization is applied:

$$\tilde{w}_{i,j} = \frac{w_{i,j}}{\sqrt{\sum_{k=1}^{n} w_{k,j}^2}}.$$  (8)

2. Ideal Solutions: The hypothetical ideal solution $Z^+$ and negative-ideal solution $Z^-$ are defined as:

$$Z^+ = [\max(\tilde{W}_{\text{DON}}), \min(\tilde{W}_{\text{NOD}})]$$  (9)

$$Z^- = [\min(\tilde{W}_{\text{DON}}), \max(\tilde{W}_{\text{NOD}})]$$  (10)

These represent the hypothetical "best" and "worst" cases, where DON is maximized and NOD minimized (for $Z^+$), and vice versa (for $Z^-$).

3. Distance Calculation: The Euclidean distance of each sample to $Z^+$ and $Z^-$ is computed:

$$D_i^+ = \sqrt{\sum_{j=1}^{2} (\tilde{w}_{i,j} - Z_j^+)^2}$$  (11)

$$D_i^- = \sqrt{\sum_{j=1}^{2} (\tilde{w}_{i,j} - Z_j^-)^2}$$  (12)

4. TOPSIS Score: The final score for each sample is:

$$C_i = \frac{D_i^-}{D_i^+ + D_i^-},$$  (13)

where $C_i \in [0, 1]$. Higher scores indicate closer alignment with $Z^+$ and greater divergence from $Z^-$.

## G  TRAINING SETTINGS

- Global batch size: 16
- Micro-batch size: 1
- Learning rate: 2e-5
- Optimizer: AdamW (Loshchilov & Hutter, 2019)
- Warmup-ratio 0.025
- Every model is trained on 8 NVIDIA A100 GPUs for less than 1 hour

## H  BAD SAMPLES GALLERY

How many edges does a cube have?
I need to identify how many edges a cube has. I remember that a
    cube is a three−dimensional shape with six square faces.Each
    square face has four edges, but when considering edges that
    are shared between two faces, I can't count all edges
    separately.A cube has 12 unique edges. They can be grouped
    into three sets of four edges that run parallel to each of the
    cube's three dimensions.Thus, after considering all faces and
    avoiding double counting, I confirm that the total number of
    edges in a cube is 12.\n Therefore the final answer is A cube
    has 12 edges.

Find the number of vertices of the rectangular prism.

I need to find the number of vertices of a rectangular prism. I know that a rectangular prism is a 3D shape with rectangular faces, length, width, and height. I can imagine that each corner of the prism is a vertex, and there are 2 corners at each edge. I can count 4 vertices at the top face, 4 vertices at the bottom face, and no additional vertices at the edges, since each edge has 2 vertices already counted. I conclude that there are 4 + 4 = 8 vertices in a rectangular prism.\n Therefore the final answer is A rectangular prism has 8 vertices.

If you select three of the four numbers 1, 2, 3, and 4, allowing duplicates, and arrange them in a row to form a three-digit natural number. How many multiples of 3 are there?\n
Therefore the final answer is There are 22 multiples of 3 that can be formed by selecting three of the four numbers 1, 2, 3, and 4 (allowing duplicates) and arranging them in a row to form a three-digit number.

What is the smallest number among the three numbers 5, 8, and 4?
I am asked to find the smallest number among three given numbers: 5, 8, and 4. I will compare the three numbers by looking at their values. I can see that 4 is the smallest number among the three, as it has the lowest value.\n Therefore the final answer is The smallest number among 5, 8, and 4 is 4.

The function $f$ is defined as follows: \n$$f(x) = 4 + \frac{\\ \text{sqrt}}{x}.$$ \nWhat is the value of $f(144)$?\n\nA. 0 \nB. 16 \nC. 40 \nD. 76
We start by substituting 144 into the function f(x). The function is defined as: f(x) = 4 + sqrt(x). So we need to calculate f(144). Substituting 144 into the function, we have: f(144) = 4 + sqrt(144). Next, we calculate sqrt(144). The square root of 144 is 12. Now we can substitute this back into the function: f(144) = 4 + 12. Adding these values together gives: f(144) = 16.\n Therefore the final answer is 16

You subtracted 4.3 when you should have added 4.3 to a number, resulting in 8.8. Find the correct value.
\n Therefore the final answer is The correct value is 17.4.

Karen bakes 50 chocolate chip cookies. She keeps 10 for herself, and she gives 8 to her grandparents. Karen wants to give everyone in her class cookies. Each person in her class will receive 2 cookies. How many people are in Karen's class?
Therefore the final answer is Karen has 16 people in her class.

Ronnie is a train engineer. His train pulls 3 red boxcars, 4 blue boxcars, and 7 black boxcars. The black boxcars can each hold 4000 pounds of coal, and the red boxcars can hold 3 times as much as the blue boxcars. All of the train's boxcars combined can hold 132,000 pounds of coal. What is the ratio of the coal capacity of each black boxcar to each blue boxcar?
I understand that Ronnie's train has 3 red boxcars, 4 blue boxcars, and 7 black boxcars, and we need to find the ratio of the coal capacity of each black boxcar to each blue boxcar. I recognize that the black boxcars can hold 4000 pounds of coal, and the red boxcars can hold 3 times as much as the blue boxcars, and the total capacity of all boxcars is 132,000

pounds. I will let the coal capacity of each blue boxcar be B pounds, so each red boxcar can hold 3B pounds. I can write an equation based on the total capacity of all boxcars: 3(3B) + 4 B + 7(4000) = 132,000. I simplify the equation by combining like terms: 9B + 4B + 28,000 = 132,000. I combine like terms to get: 13B + 28,000 = 132,000. I subtract 28,000 from both sides to get: 13B = 104,000, and then divide both sides by 13 to get: B = 8,000. I know that each blue boxcar can hold 8,000 pounds, and each black boxcar can hold 4,000 pounds, so the ratio of the coal capacity of each black boxcar to each blue boxcar is 4000/8000 = 1/2.
\n Therefore the final answer is The ratio is 1:2.

## I EMPIRICAL VALIDATIONS OF OUTPUT LAYER FOCUS

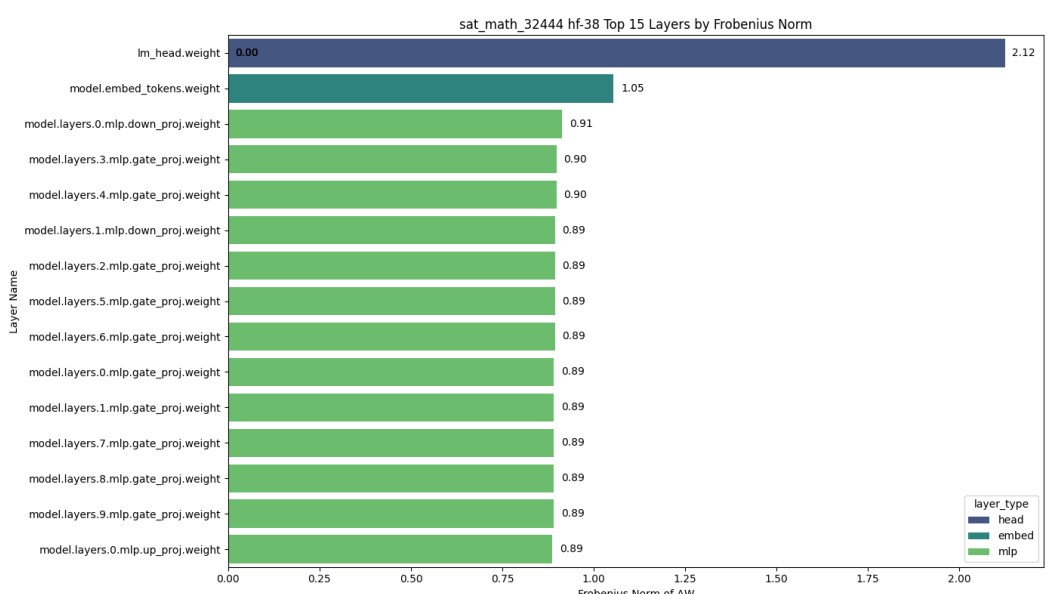

Figure 6: Ranking of Frobenius norm delta of layers of Llama-3.1-8B-Instruct after fine-tuning on SAT Math COT dataset, epoch 2

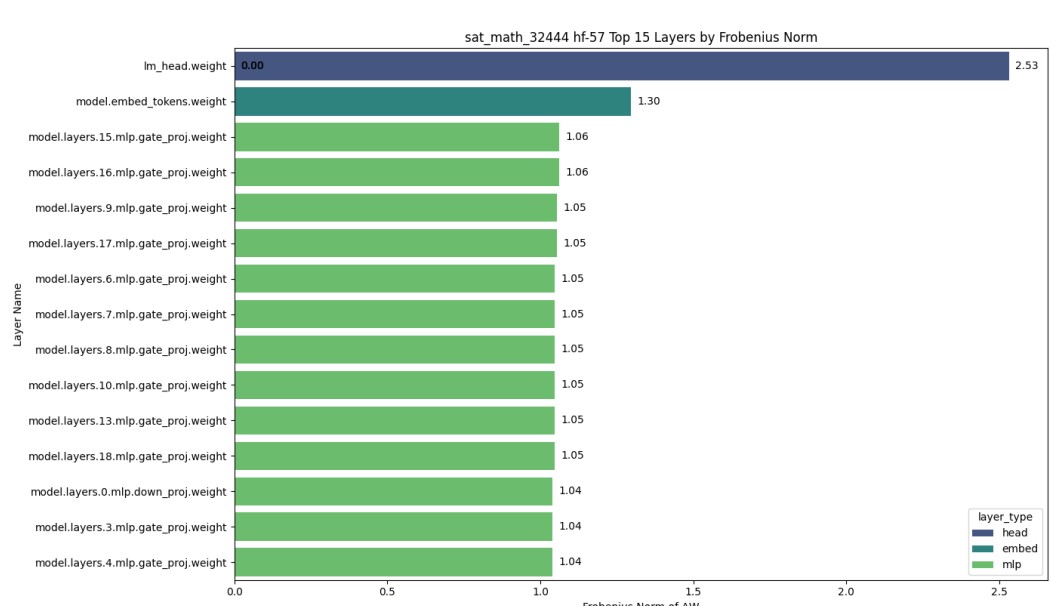

Figure 7: Ranking of Frobenius norm delta of layers of Llama-3.1-8B-Instruct after fine-tuning on SAT Math COT dataset, epoch 3

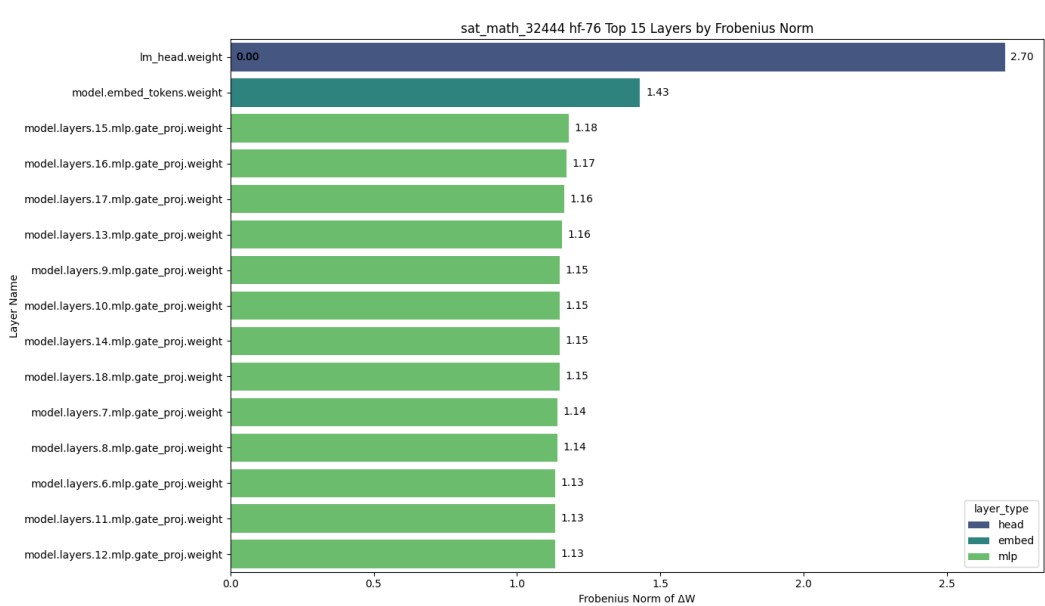

Figure 8: Ranking of Frobenius norm delta of layers of Llama-3.1-8B-Instruct after fine-tuning on SAT Math COT dataset, epoch 4

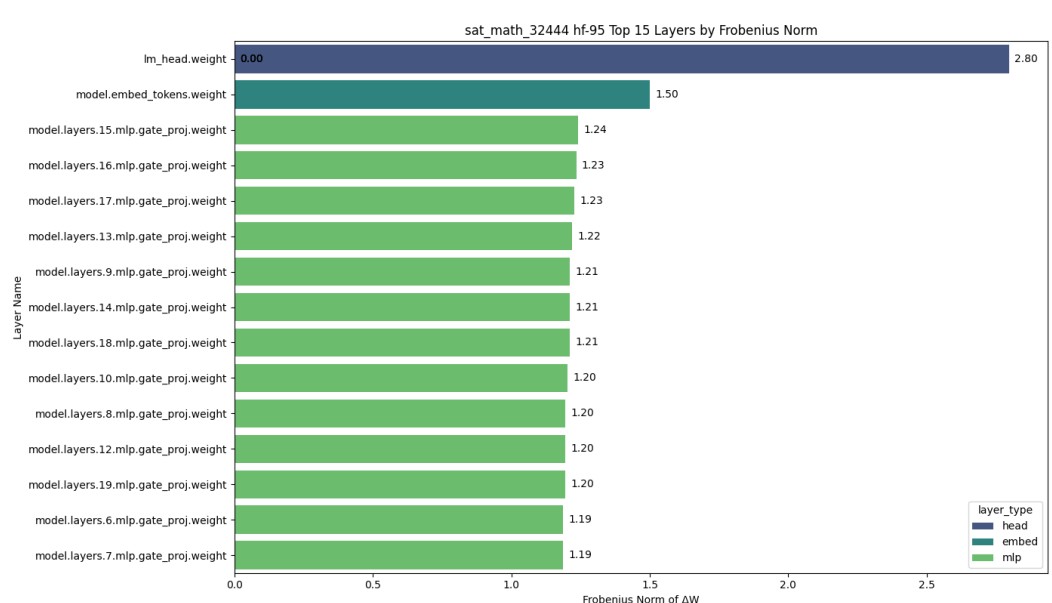

Figure 9: Ranking of Frobenius norm delta of layers of Llama-3.1-8B-Instruct after fine-tuning on SAT Math COT dataset, epoch 5

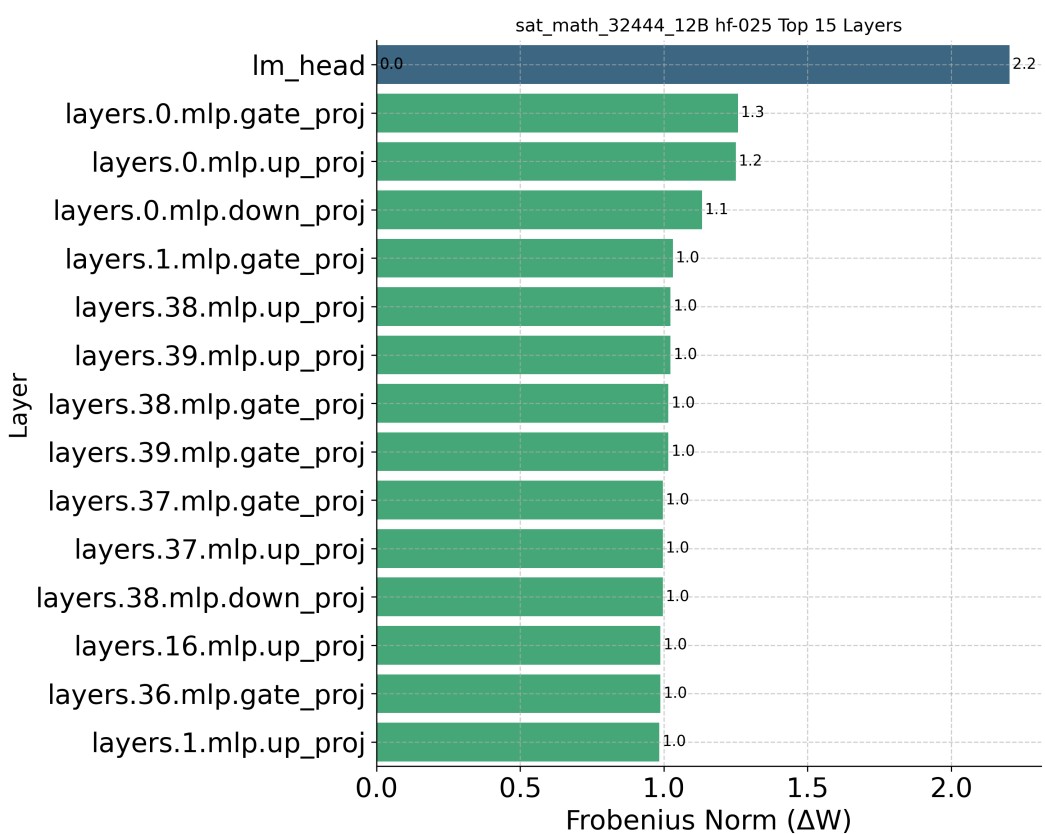

Figure 10: Ranking of Frobenius norm delta of layers of Llama-2-13b-chat after fine-tuning on SAT Math COT dataset, epoch 1

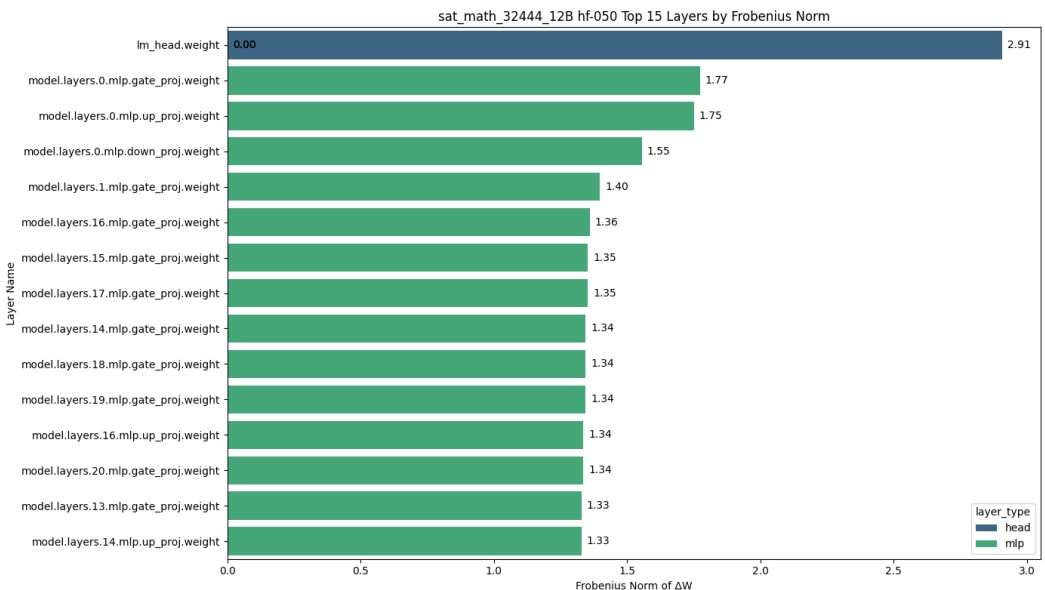

Figure 11: Ranking of Frobenius norm delta of layers of Llama-2-13b-chat after fine-tuning on SAT Math COT dataset, epoch 2

## J    BENCHMARK DETAILS

Table 5: Constructed Benchmark Details. To ensure the representativity of each measured ability, multiple datasets focusing on the same ability are selected, mitigating potential biases arising from the limited scope of individual exams. Additionally, the inclusion of the comprehensive and unrelated dataset IFEval-en further enhances the reliability and representative of the benchmark, aligning it more closely with real-world use cases.

| Dataset | Ability | Validation Set Available | Test Set Size |
|---------|---------|--------------------------|---------------|
| LSAT-AR | Logical Reasoning | False | 230 |
| LSAT-LR | Logical Reasoning | False | 510 |
| LogiQA-en | Logical Reasoning | True | 651 |
| LSAT-RC | Reading Comprehension | False | 269 |
| SAT-en | Reading Comprehension | False | 206 |
| AQUA-RAT | Mathematical Problem-Solving | True | 254 |
| SAT-math | Mathematical Problem-Solving | False | 220 |
| IFEval-en | Instruction Following | False | 541 |

## K    DATASETS DETAILS

Table 6: Details of the datasets used in the study.

| Dataset | Ability | Label Type | Size |
|---------|---------|------------|------|
| LogiQA Train | Logic Reasoning | Single letter only | 7,851 |
| SAT Math COT | Mathematical Problem-Solving | COT | 32,444 |
| GSM8K | Mathematical Problem-Solving | COT | 8794 |
| Ifeval-Like Data | Instruction Following | General text | 56.3K |

## L    ABLATION CONFIGURATIONS

Table 7: Ablation study configurations.

| Configuration | DON | NOD | TOPSIS |
|---------------|-----|-----|--------|
| DON Only | ✓ | ✗ | ✗ |
| NOD Only | ✗ | ✓ | ✗ |
| DON + NOD (Weighted Sum) | ✓ | ✓ | ✗ |
| DON + NOD (Pareto Front) | ✓ | ✓ | ✗ |
| DON + NOD + TOPSIS (Full) | ✓ | ✓ | ✓ |

