# OpenReview forum: "DONOD: Efficient and Generalizable Instruction Fine-Tuning for LLMs via Model-Intrinsic Data Selection"
_ICLR.cc/2026/Conference — Submitted to ICLR 2026_

### Official Review · Reviewer_jtvs · 2025-10-23

**Soundness:** 2
**Presentation:** 3
**Contribution:** 2
**Rating:** 4
**Confidence:** 5

**Summary:**

This paper introduces a model-intrinsic data selection method designed to enhance the efficiency and generalization of instruction tuning for llms. The authors propose two metrics, Delta of Norm (DON) and Norm of Delta (NOD), to evaluate the impact of training samples on model weights. By integrating these metrics with the TOPSIS algorithm, the method filters out noisy, unlearnable, and generalization-harming samples without relying on auxiliary models or validation sets. Experiments demonstrate that the proposed method help improve both target-domain and cross-domain accuracy by selecting a subset of the training data.

**Strengths:**

- The paper studies how targeted instruction tuning can be performed without significantly degrading generalization to other tasks, which is an important aspect that is ignored in prior works.
- The method is conceptually simple, easy to implement in practice, and generalizable across different model architectures.
- The paper includes detailed ablation studies to validate the contributions of individual components (DON, NOD, and TOPSIS).

**Weaknesses:**

- The paper appears to conflate two distinct problems in data selection. The first is targeted data selection, which focuses on identifying a domain-specific subset from a large, heterogeneous data corpus (works such as [1][2][3] focus on this problem). The second is within-domain selection or pruning, which refines an already domain-related dataset (works such as [4][5][6] focus on this problem). Without a clearly defined reference or validation dataset, it is not possible to address the first problem effectively, and the paper’s framing seems to overlook this distinction.
- The evaluation is not solid:
  - The baselines are weak. The experiments only compare the proposed method with LESS, ALL and RANDOM. LESS is focusing on targeted data selection, which is not a good baseline in this paper's settings. It would be more convincing if the paper compares with stronger data selection or pruning methods such as [5] and [6].
  - The experimental results are reported without standard deviations or confidence intervals, which makes it difficult to assess the statistical significance and robustness of the findings.
  - Experiments such as "ROBUSTNESS IN IDENTIFYING NOISE" do not compare with any baselines, making it impossible to tell the quality of noise filtering.

[1] Sang Michael Xie, Shibani Santurkar, Tengyu Ma, and Percy Liang. 2023. Data selection for language models via importance resampling. In Proceedings of the 37th International Conference on Neural Information Processing Systems (NIPS '23). Curran Associates Inc., Red Hook, NY, USA, Article 1482, 34201–34227.

[2] Mengzhou Xia, Sadhika Malladi, Suchin Gururangan, Sanjeev Arora, and Danqi Chen. 2024. LESS: selecting influential data for targeted instruction tuning. In Proceedings of the 41st International Conference on Machine Learning (ICML'24), Vol. 235. JMLR.org, Article 2221, 54104–54132.

[3] Zifan Liu, Amin Karbasi, and Theodoros Rekatsinas. 2025. TSDS: data selection for task-specific model finetuning. In Proceedings of the 38th International Conference on Neural Information Processing Systems (NIPS '24), Vol. 37. Curran Associates Inc., Red Hook, NY, USA, Article 325, 10117–10147.

[4] Coleman, Cody, et al. "Selection via proxy: Efficient data selection for deep learning." arXiv preprint arXiv:1906.11829 (2019).

[5] Tan, Haoru, et al. "Data pruning by information maximization." arXiv preprint arXiv:2506.01701 (2025).

[6] Cho, Y., Shin, B., Kang, C. &amp; Yun, C.. (2025). Lightweight Dataset Pruning without Full Training via Example Difficulty and Prediction Uncertainty. Proceedings of the 42nd International Conference on Machine Learning, in Proceedings of Machine Learning Research 267:10602-10643

**Questions:**

- Could you clarify what is the selection process in the experiments for both the proposed method and LESS? What are the candidates? How the validation set is used if exists?
- In Table 2, what causes the 30% selection to perform worse than the 20% selection? If performance is not monotonic with respect to the selection ratio, how should the threshold be chosen?
- How can the Frobenius norm capture fine-grained structural changes, given that it represents an aggregated measure?

---

### Official Review · Reviewer_vBWj · 2025-10-26

**Soundness:** 2
**Presentation:** 3
**Contribution:** 2
**Rating:** 4
**Confidence:** 4

**Summary:**

The paper introduces two TOPSIS-balanced influence metrics for data selection, aiming to optimize for both target-domain efficiency and non-target generalization. While results show some improvement, they are not convincing due to limited comparison against other methods and anlysis.

**Strengths:**

The paper is well-motivated, explicitly addressing the important trade-off between target-domain specialist and non-target generalist performance in data curation.

**Weaknesses:**

1. The paper's claims are unconvincing as it fails to do comprehensive survey and compare against relevant SOTA data selection methods. For example, but not limited to:
- What Makes Good Data for Alignment? A Comprehensive Study of Automatic Data Selection in Instruction Tuning
- DataMan: Data Manager for Pre-training Large Language Models
-IMPROVING DATA EFFICIENCY VIA CURATING LLM-DRIVEN RATING SYSTEMS
- SelectIT: Selective Instruction Tuning for LLMs via Uncertainty-Aware Self-Reflection

The paper's claims would be far more convincing if benchmarked against these and other modern data curation strategies.

2. The norm-based influence score is a simplistic proxy for generalization. More importantly, the score-based selection method completely ignores data diversity, a critical factor for robust fine-tuning.

3. As shown in Figure 4, the method is highly sensitive to the budget setting, undermining its practical utility. The motivation and method formulation offer no explanation for the observations (e.g., why performance peaks at small budgets for some datasets).

**Questions:**

1. What is the generalization performance of two crucial baselines: (a) a simple data mixture of all target domains, and (b) the base instruction-tuned model with no downstream fine-tuning at all?

2. The setting is established upon instruction-tuned models, eg LLaMA-3.2-3B-Instruct. Did the authors check for and clean potential data leakage from the evaluation benchmarks, which is a common issue?

---

### Official Review · Reviewer_burq · 2025-10-29

**Soundness:** 2
**Presentation:** 2
**Contribution:** 2
**Rating:** 4
**Confidence:** 4

**Summary:**

The paper proposes a data selection method for instruction fine-tuning of large language models, named DONOD. This method leverages changes in model weights for selection. One metric, Delta of Norm, measures the change of the $\ell_2$-norm in model weights before and after fine-tuning on a single sample. Another metric, Norm of Delta, measures the $\ell_2$-norm of the weight difference before and after fine-tuning on a single sample. Based on these metrics, a multi-criteria decision method is used to maximize DON and minimize NOD. Only the output layer weights are used to approximate model behavior.

The experiments evaluate DONOD on instruction-tuning datasets, including GSM8K, SAT Math Chain-of-Thought, LogiQA, and IFEval-like data. Models including LLaMA-3.1-8B-Instruct, LLaMA-2-13B-Chat, LLaMA-3.2-3B-Instruct, and Qwen-2.5-7B-Instruct are fine-tuned using subsets of these datasets selected by DONOD. The baselines are full-dataset supervised fine-tuning (ALL), random sampling (Random), and existing data selection methods such as LESS (influential data selection) and IFD (influence-driven filtering). Across all datasets, DONOD surpasses the baselines while using 20–30% of the training data, achieving up to 14.9% improvements in target-domain accuracy and 5.67% in cross-domain generalization compared to full-data training. It exhibits cross-architecture transferability, as data selected with smaller models generalize effectively to larger ones. The experiments show robustness to noisy data by filtering corrupted or mislabeled samples.

**Strengths:**

- This paper introduces a data selection approach that uses training dynamics (via the DON and NOD metrics) without external reward models or validation sets. By computing on the output layer, it reduces computational costs.

- The method outperforms full-data fine-tuning and established baselines such as LESS, IFD, and random sampling across diverse benchmarks (GSM8K, SAT Math COT, LogiQA, and IFEval).

- DONOD exhibits robustness in the presence of corrupted or mislabeled samples. Under label noise, the method successfully filters out harmful data and maintains performance stability.

**Weaknesses:**

- The motivation for using the two metrics is not clear. Are there intuitive examples showing that the two metrics' correlation with the generalization performance? Or is there any theoretical guarantee or formulation? In other words, why does this method perform better than a gradient-similarity-based method, like LESS?

- How this method computes the metrics is not clear. It measures the change in the model weights before and after fine-tuning on a single sample. How are the metrics evaluated? Is it by fine-tuning one model on each sample, which may be prone to overfitting?

- It would be better to provide an algorithm box in the paper.

- It would be better to provide a runtime comparison with existing works.

**Questions:**

- What is the training dataset used in this paper?
- Why only compute the two metrics in this paper in the last layer? It would be better to conduct an ablation study on various types of layers.
- What is the interplay when choosing based on the weighted combination of the DON and NOD metrics.

---

### Official Review · Reviewer_b8oK · 2025-11-02

**Soundness:** 2
**Presentation:** 3
**Contribution:** 2
**Rating:** 4
**Confidence:** 2

**Summary:**

The paper proposes DONOD, a model-intrinsic data selection framework for instruction fine tuning of LLMs. Each training sample is scored by two parameter-space metrics computed from a one-step update: Delta of Norm (DON), the change in Frobenius norm of weights, intended as a proxy for generalization (prefer higher positive DON), and Norm of Delta (NOD), the Frobenius distance between pre/post weights, intended to penalize unstable/noisy samples (prefer lower NOD). The two criteria are combined via TOPSIS to rank and retain a subset for SFT. The authors argue this avoids auxiliary models and validation sets, is lightweight via focusing on the output layer, and improves both in-domain and cross-domain performance on math/logic/reading/instruction benchmarks. Reported results show that pruning 70- 80% of data can outperform full-data SFT and that subsets selected on one model transfer to others.

**Strengths:**

1. The proposed new metrics (DON and NOD) are simple to compute (Frobenius norms) and require no external judge or reward model.
2. The proposed method is efficient as the computation can be focused on the latter layers, especially the output layer.
3. The experimental results look promising.

**Weaknesses:**

1. The proposed metrics are computed by comparing model weights **before** and **after** SFT, but it is unclear how the contribution of a single sample is isolated. If the method relies on a one-step “probe” update per sample, the exact training protocol must be specified.

2. If the samples are individually evaluated (one sample by one sample), this method may ignore interactions among examples (e.g., redundancy, curriculum effects, and gradient conflicts). As a result, a set built from top-ranked individual scores may not be optimal as a subset.

3. The paper alternates between describing a “ample” and a “fine-tuning dataset” when discussing the metrics. It creates ambiguity about whether DON/NOD are intended as metrics for per-sample or a aggregated subset.

**Questions:**

NA

---

### Meta-Review · Area_Chair_aG74 · 2026-01-07

**Summary:**

The paper proposes DONOD, a lightweight, model-intrinsic data selection method for instruction fine-tuning of LLMs, using two parameter-norm-based metrics (DON and NOD) combined via TOPSIS to prune training data. While the method reports strong empirical gains—e.g., improved in-domain and cross-domain accuracy with 70–80% less data—reviewers raise consistent concerns about theoretical motivation, experimental rigor, and comparison breadth, which collectively weaken confidence in the claims.

**Reviewer Concerns:**

No rebuttal was submitted, so none of the concerns were addressed. All major issues remain outstanding: (1) lack of clear justification for why DON/NOD correlate with generalization; (2) insufficient comparison to recent, relevant data selection/pruning baselines; (3) ambiguity in metric computation (e.g., per-sample updates, potential overfitting); (4) missing statistical error bars and non-monotonic performance trends without explanation; and (5) neglect of data diversity and limited ablation studies (e.g., on layer choice or metric weighting).

**Reviewer Scores:**

All four reviewers gave a rating of 4 (“marginally below acceptance threshold”) with confidence levels ranging from 2 to 5. In the absence of a rebuttal and given unresolved concerns, none would likely raise their score to a clear accept (≥6). Scores would probably remain at 4 or possibly drop to 3 if deeper flaws were discussed.

---

### Decision · Program_Chairs · 2026-01-26

Reject